# Hydrocortisone, Ascorbic Acid and Thiamine (HAT Therapy) for the Treatment of Sepsis. Focus on Ascorbic Acid

**DOI:** 10.3390/nu10111762

**Published:** 2018-11-14

**Authors:** Paul E. Marik

**Affiliations:** Division of Pulmonary and Critical Care Medicine, Eastern Virginia Medical School, Norfolk, VA 23507, USA; marikpe@evms.edu

**Keywords:** sepsis, septic shock, hydrocortisone, vitamin C, ascorbic acid, thiamine

## Abstract

Sepsis is a devastating disease that carries an enormous toll in terms of human suffering and lives lost. Over 100 novel pharmacologic agents that targeted specific molecules or pathways have failed to improve the outcome of sepsis. Preliminary data suggests that the combination of Hydrocortisone, Ascorbic Acid and Thiamine (HAT therapy) may reduce organ failure and mortality in patients with sepsis and septic shock. HAT therapy is based on the concept that a combination of readily available, safe and cheap agents, which target multiple components of the host’s response to an infectious agent, will synergistically restore the dysregulated immune response and thereby prevent organ failure and death. This paper reviews the rationale for HAT therapy with a focus on vitamin C.

## 1. Introduction

The global burden of sepsis is substantial with an estimated 32 million cases and 5.3 million deaths per year per year with most of these cases occurring in low-income countries [1]. Data from the U.S. and Australia demonstrates that over the last two decades the annual incidence of sepsis has increased by approximately 13% with a decrease in in-hospital mortality from about 35% to 20% [2,3,4]. In 2013, over 1.3 million patients were hospitalized in the U.S. with a diagnosis of sepsis, of which over 300,000 died [5]. In addition to short-term mortality, septic patients suffer from numerous long-term complications with a reduced quality of life and an increased risk of death up to five years following the acute event [6,7,8,9,10]. Approximately 50% of sepsis survivors develop post-sepsis syndrome characterized by the development of new psychiatric and cognitive deficits [11]. Post sepsis-syndrome is similar in many respects to post-traumatic stress disorder (PTSD); patients suffer memory impairment, abnormalities of higher executive function, nightmares, anxiety disorders and depression [12]. Apart from the enormous financial costs of sepsis, the human toll of this disease is staggering and new interventions that limit the ravages of this disease are urgently required. 

In patients with sepsis organ failure and death, it is usually a result of the host’s response to the infecting pathogen rather than from the infecting pathogen itself. This was first recognized by Sir William Osler, who commented that “*except on few occasions, the patient appears to die from the body’s response to infection rather than from the infection*” [13]. Sepsis is fundamentally an inflammatory disease mediated by the activation of the innate immune system by both pathogen-associated molecular patterns (PAMPs) and damage-associated molecular patterns (DAMPs). Calvano et al. demonstrated that exposure of blood leukocytes to bacterial endotoxin (LPS) altered the expression of 3714 genes [14]. These include genes for pro- and anti-inflammatory cytokines, chemokines, adhesion molecules, transcription factors, enzymes, clotting factors, stress proteins and anti-apoptotic molecules [15]. These inflammatory mediators have widespread pathophysiologic consequences, including vasoplegic shock, myocardial dysfunction, altered microvascular flow and diffuse endothelial injury [16,17]. However, fundamentally, sepsis is characterized by the excessive production of reactive oxygen species (ROS) by the induction of enzymes such as nicotinamide adenine dinucleotide phosphate-oxidase (NOX) and the uncoupling of mitochondrial oxidative phosphorylation (see Figure 1) [18]. In addition, ROS are produced by xanthine oxidase, lipoxygenase and cyclooxygenase. Important ROS in sepsis pathogenesis includes hydrogen peroxide (H_2_O_2_), superoxide (O_2_^−^), hydroxyl radicals (HO^·^), peroxynitrite (ONOO^−^) and hypochlorous acid (HOCl).

The excess production of ROS underlie many of the pathological processes characteristic of sepsis [20]. ROS has been shown to modulate the lipopolysaccharide-Toll like receptor 4 (LPS-TLR4) signaling pathway. ROS activate nuclear factor kappa-B (NF-κB) by activation of inhibitory kappa-B kinase (IκB kinase). NF-κB increase the transcription of multiple pro-inflammatory mediators (see Figure 1). When the hosts antioxidant defenses are overwhelmed, ROS can induce injury to lipids, proteins and nucleic acids, thereby resulting in widespread endothelial dysfunction, mitochondrial dysfunction, cellular injury, and multiple organ dysfunction [20]. Isoprostanes are products of lipid peroxidation that are formed non-enzymatically and can be measured in blood, urine, or tissues. Because of their stability and high specificity, the F2-isoprostanes are currently considered to be the most reliable biomarkers of in vivo oxidative stress and lipid peroxidation. In a cohort of patients with sepsis, Ware et al. demonstrated that F2-isoprostane levels were higher in patients who developed organ failure and in those who died [21]. Similarly, Kumar et al. reported augmented levels of oxidants in patients with sepsis, as demonstrated by DMPO nitrone adduct formation and plasma myeloperoxidase (MPO) level activity [22]. In this study, the SOFA and APACHE II scores correlated linearly with the MPO levels and inversely with the levels of antioxidants. This study supports the concept that the imbalance between oxidant and antioxidants plays a key role in the pathophysiology of organ failure in sepsis. 

The unbalanced production of mitochondrial ROS impairs mitochondrial structure, enzymatic function, and biogenesis. Mitochondria are both the target and source of ROS in sepsis. In an in vitro model of sepsis-induced kidney injury, Quoilin et al. demonstrated that LPS induces NADPH oxidase and inducible nitric oxide synthetase (iNOS) expression. Following this, uncoupling of the mitochondrial respiratory chain due to the inhibition of complex IV and reduction in ATP levels were observed [23]. Mitochondrial dysfunction was associated with cytochrome c release and the loss of mitochondrial membrane potential. Alterations in fatty acid metabolism with decreased beta-oxidation and abnormalities of the citric acid cycle appear to be a characteristic feature of the mitochondrial dysfunction in sepsis [24,25]. These changes lead to decreased ATP production. In the heart, myocyte oxidative injury is accompanied by increased proteolysis, mitochondrial damage, dysregulated nitric oxide metabolism, β-adrenoceptor down-regulation and calcium mishandling [26,27,28]. The unbalanced production of mitochondrial ROS impairs mitochondrial structure, enzymatic function and biogenesis, and plays a role in the metabolic failure of sepsis. These data suggest that excessive production of ROS plays a pivotal role in the pathophysiology of sepsis and that interventions that neutralize oxidants would have a protective role in sepsis. Furthermore, proinflammatory mediators increase the expression of pyruvate dehydrogenase kinase isoenzymes (PDK4), which inactivate the pyruvate dehydrogenase complex (PDC) [29,30]. Inactivation of the PDC prevents pyruvate entering the Krebs cycle potentiating the metabolic failure caused by mitochondrial dysfunction. This phenomenon is compounded by thiamine deficiency and cytokine-mediated down regulation of expression of the PDC [29]. 

Over the last three decades, over 100 phase II and phase III clinical trials have been performed testing various novel pharmacologic agents and therapeutic interventions in an attempt to improve the outcome of patients with severe sepsis and septic shock; all of these efforts ultimately failed to produce a novel pharmacologic agent that reduced organ failure and improved the survival of patients with sepsis [31]. All these studies used a single agent that targeted a specific molecule or pathway; due to the thousands of mediators involved and the redundancy of the multiple pathways, such an approach was doomed to fail. Furthermore, the central role of ROS was ignored. Dr Aird commented that “*the best hope for therapeutic advances [in sepsis] will depend on broad-base targeting, in which multiple components are targeted at the same time*” [32]. We believe that the combination of Hydrocortisone, Ascorbic Acid and Thiamine (HAT therapy) achieves these goals. The premise behind HAT therapy is the use of a combination of readily available, safe and cheap agents that target multiple components of the host’s response to an infectious agent such that they synergistically restore the dysregulated host immune response, neutralize damaging oxidants, and restore mitochondrial function. We believe this unique and novel approach has the potential to reduce the global burden of sepsis, reduce the post-sepsis syndrome without side-effects, and be highly cost-effective. A cost analysis indicates that HAT therapy has the potential to save billions of dollars and millions of life-years in the United States [33]. This paper reviews the rationale for HAT therapy with a focus on vitamin C.

## 2. Vitamin C

It has been known for over 20 years that in acutely-ill patients [34,35,36] as well as in experimental models of sepsis that an acute deficiency of vitamin C develops, characterized by low serum and intracellular levels of the vitamin [37,38,39,40,41]. Critically ill septic patients typically have very low or undetectable serum levels of vitamin C, resulting in an acute scorbutic condition [35,42,43,44]. Recently Carr and co-authors demonstrated that 100% of septic patients had low vitamin C levels, 88% had hypovitaminosis C (<23 μmol/L), while 38% had a severe deficiency (<11 μmol/L) [45]. Low vitamin C levels in critically ill patients are associated with increased vasopressor requirements, kidney injury, multiple organ dysfunction (higher SOFA scores) and increased mortality [35,44]. The underlying cause of vitamin C deficiency is likely due to increased oxidation (metabolic consumption), decreased absorption, and increased urinary losses of the vitamin. In a murine caecal-ligation and perforation model (CLP), Armour et al. reported that the plasma ascorbate level fell rapidly by 50% and this was associated with a 1000% increase in the urine ascorbate concentration [38]. Sepsis induced glomerular hyperfiltration and/or tubular dysfunction results in decreased tubular reabsorption of filtered vitamin C with increased urinary losses [46].

Vitamin C is a potent antioxidant, which directly scavenges oxygen free radicals, restores other cellular antioxidants including tetrahydrobiopterin and α-tocopherol and is an essential co-factor for iron and copper containing enzymes [18,47]. Vitamin C is a key cellular antioxidant, detoxifying exogenous oxidants radical species that have entered cells or which have arisen within cells due to excess superoxide generation by mitochondrial metabolism, by NADPH oxidase, xanthine oxidase or by uncoupled nitric oxide synthase (NOS) [18,48]. The low electron reduction potential of both vitamin C and its one-electron oxidation product, the ascorbyl radical, enable them to reduce most clinically important radicals and oxidants.

Dehydroascorbic acid, the two-electron oxidation product of ascorbic acid, is transported via the GLUT1 transporter into mitochondria, where it converted to ascorbic acid and acts as a potent antioxidant limiting mitochondrial oxidant injury [49,50]. Considering that the mitochondrial respiratory chain is a main source of ROS in live cells and mitochondrial dysfunction plays a prominent role in sepsis pathogenesis, antioxidants targeting the intra-mitochondrial environment could be pivotal role in the treatment of sepsis. Furthermore, ascorbic acid is required for the synthesis of carnitine, which is required for the transport of fatty acids into the mitochondrial matrix and for beta-oxidation [24,49]. Carnitine deficiency may occur in the context of sepsis and preliminary data suggests that an infusion of l-carnitine may be beneficial in patients with septic shock [51]. In an experimental model, Dhar-Mascareno and colleagues demonstrated that oxidant induced mitochondrial damage and apoptosis in human endothelial cells were inhibited by vitamin C [52]. In a CLP sepsis model, Kim et al. administered 100 mg/kg ascorbic acid immediately after sepsis induction [53]. In this study, vitamin C attenuated the elevation in serum aminotransferase and hepatic lipid peroxide levels. Studies using *N*-acetylcysteine (a synthetic anti-oxidant) have proven to be ineffective and potentially harmful in patients with sepsis, possibly due to the limited ability of this drug to enter into the mitochondria and its inability to regenerate BH4 [18,54,55].

Vitamin C suppresses activation of NF-κB by inhibiting tumor necrosis factor-α (TNFα) induced phosphorylation of inhibitory kappa-B kinase (IκB kinase) [56]. Ascorbic acid decreases high mobility group box 1(HMGB1) secretion [57]; HMGB1 is an important late pro-inflammatory cytokine. Vitamin C may decrease the synthesis and inactivate histamine [58]; histamine has been shown to play an important role in sepsis [59]. Vitamin C is an essential co-factor for the synthesis of norepinephrine, epinephrine and vasopressin; in addition vitamin C increases adrenergic transmission [60]. Vitamin C may decrease the immunosuppression associated with sepsis. It has been known for over 60 years that vitamin C has immune-enhancing properties. In 1949, Dr Fred Klenner from Reidsville, North Carolina, reported on the use of intravenous vitamin C in the treatment of polio and other viral illnesses [61]. It was initially assumed that vitamin C was directly viricidal (in vivo) and this mistaken belief underlies the recommendations of Linus Pauling who promoted the use of large doses of oral vitamin C (up to 18 g/day) for the prevention and treatment of the common cold [62]. A number of RCTs have reported that vitamin C supplementation had no effect on the incidence of the common cold [63]. However, vitamin C has been shown to decrease the incidence of the common cold if the person is under enhanced stress, e.g., cold temperatures and/or physical stress [64]. While high dose vitamin C has in-vitro viricidal properties [65,66], there is no data or physiologic rationale to suggest that this occurs in vivo. Rather, the “anti-viral” effect of vitamin C are likely due to that fact that vitamin C has specific immune-enhancing effects. Vitamin C is concentrated in leucocytes, lymphocytes and macrophages, reaching high concentrations in these cells [67]. Vitamin C improves chemotaxis, enhances neutrophil phagocytic capacity and oxidative killing, stimulates interferon production, and supports lymphocyte proliferation [68,69,70]. The major presumed beneficial effects of vitamin C in patients with sepsis are outlined in Table 1. 

While vitamin C is the most potent and important anti-oxidant in mammals, in the presence of transition metals (iron and copper), vitamin C may paradoxically be associated with a pro-oxidant effect [71]. In the presence of free iron, vitamin C may reduce free ferric iron to the ferrous form. The ferrous form then undergoes a Fenton-type reaction with hydrogen peroxide yielding hydroxyl or hydroxyl-like reactions. Vitamin C may generate ROS in in-vitro, cell culture or tissue incubation experiments, where free metal ions might exist [71,72,73]. Normally, iron is tightly bound to protein and does not exist in the free form. However, in conditions such as hypoxia (ischemia/reperfusion) and sepsis, free iron may be released from ferritin. Furthermore, sepsis is associated with hemolysis and the release of free heme. Free heme can be highly cytotoxic in the presence of proinflammatory mediators [74,75]. The divalent iron atom contained within its protoporphyrin IX ring can promote the production of free radicals. In the presence of free iron or free heme and depending on the dose of vitamin C and the timing of the administration of the dose in relationship to inciting event, vitamin C may either act as an anti-oxidant or pro-oxidant. This concept was elegantly demonstrated in a hepatic ischemia/reperfusion model where Seo and Lee demonstrated that an infusion of 30 mg/kg and 100 mg/kg of vitamin C decreased markers of oxidative injury, whereas these markers were increased with a dose of 300 mg/kg and 1000 mg/kg [76]. Similarly, in a hepatic ischemia/reperfusion model, Park and colleagues demonstrated that oxidative injury was attenuated at ascorbic acid concentrations of 0.25 mM and 0.5 mM, however they were augmented at a concentration of 2 mM [77]. This concept is further supported by a number of studies using a cardiac arrest ischemia/reperfusion model. In two murine studies, vitamin C in a dose of 50 mg/kg and 100 mg/kg IV decreased myocardial damage, improved neurological outcome and the survival rate [68,69]. However, using a similar model, Motl and colleagues reported that a 250 mg/kg dose was harmful [78]. In a murine CLP model, Tyml et al. demonstrated that the delayed (24 h) administration of vitamin C (in a dose of 76 mg/kg) restored blood pressure and microvascular perfusion [41]. Using human umbilical vein endothelial cells Kuck and colleagues demonstrated that cell free hemoglobin increased endothelial permeability in part through depletion of intracellular ascorbate [79]. In this study the addition of ascorbate up to a concentration of 60 μmol/L (physiologic concentrations) attenuated the increase in permeability mediated by cell free hemoglobin [79]. This study reaffirms that low dose vitamin C may reduce the toxicity of cell free hemoglobin. 

In the absence of free iron or heme, vitamin C acts as an oxidant only in extremely high pharmacologic doses (>100 g) when used as adjunctive treatment in patients with cancer [80,81,82,83]. Repeated intravenous injection of 750–7500 mg/day of vitamin C for six days in healthy volunteers did not induce a pro-oxidant change in plasma markers [84]. Furthermore, intravenous infusion of very high-dose ascorbate (1584 mg/kg over 24 h) lowered serum malondialdehyde concentration (a marker of oxidative stress) in severely burned patients [85]. When used prophylactically (before the inciting event) prior to the oxidant injury with release of free iron, it is likely that a lower dose of vitamin C may act as a powerful antioxidant. Basili et al. demonstrated that a single 1 g infusion of vitamin C prior to the performance of percutaneous coronory reperfusion significantly improved microcirculatory perfusion with a marked reduction in the markers of oxidant injury [86]. Similarly, Wang and colleagues demonstrated that 3 g vitamin C IV prior to elective percutaneous coronory reperfusion was associated with less myocardial injury [87]. A metaanalysis, which included 8 randomized controlled trials, demonstrated that vitamin C up to a dose of 2 g IV given pre-operatively significantly reduced the risk of atrial fibrillation in patients undergoing cardiac surgery [88]. Oxidant injury is postulated to play a role in contrast-induced renal dysfunction. A systematic review by Xu and colleagues demonstrated that prophylactic vitamin C (oral and IV up to a total dose of 7 g) significantly reduced the risk of renal impairment [89].

These data suggest that in the setting of sepsis and ischemia/reperfusion, a narrow dose response curve exists. In preliminary observational data in patients with severe sepsis and septic shock and using procalcitonin as a biomarker, we have similarly observed a narrow dose response curve; the optimal daily dose appears to be approximately 6 g/day with an attenuated effect at a dose of 2 g and 10 g, respectively (personal observations). In treating over 800 patients with a 6 g/day dose, we are unaware of any patient who had a pro-oxidant effect (as reflected by the procalcitonin trajectory), even when the treatment was delayed. It is possible that delayed treatment with a higher dose may have a pro-oxidant deleterious effect. It should be recognized that there is a delicate balance between protective oxidant signaling and the detrimental effects of ROS. Additional studies are required to further elucidate the dose response of vitamin C in various clinical situations. Furthermore, we endorse the recommendation of Spoelstra-de Man and colleagues who suggest administering “*high-dose vitamin C for a short course of four days only, e.g., during the overwhelming oxidative stress when organ damage occurs. After four days, plasma concentrations will be supranormal and vitamin C can be continued in a low (nutritional) dose to allow generation of low concentrations of ROS, which are essential for physiological signaling and repair.*” [19].

## 3. Hydrocortisone

Glucocorticoids have diverse anti-inflammatory properties. These are briefly reviewed here; the reader is referred to recent publications for a more comprehensive review of this topic [90,91,92,93]. Classically, glucocorticoid binding to the glucocorticoid receptor (GR) activates or represses gene transcription, with glucocorticoids regulating up to 20% of the genome [94]. Glucocorticoids affect nearly every cell of the immune system. Glucocorticoids suppress inflammation by multiple mechanisms that impact both the innate and adaptive immune responses. The primary anti-inflammatory action of glucocorticoids is to repress a large number of pro-inflammatory genes, which encode cytokines, chemokines, inflammatory enzymes, cell adhesion molecules coagulation factors and receptors. GR-mediated repression of the transcriptional activity of NF-κB and AP-1 play a major role in mediating the anti-inflammatory actions of glucocorticoids. In addition to attenuating the pro-inflammatory response, low-dose glucocorticoids have immune-stimulating effects, which may limit the anti-inflammatory immunosuppressive state [95]. The immune-enhancing effects of glucocorticoids and the balance between the immune suppressing and enhancing effects of the drug are critically dependent on the dose and duration of treatment as well as the state of immune activation of the host.

Over the last 40 years, 22 randomized controlled trail have been conducted investigating the benefits of glucocorticoids in patients with septic shock [96]. Many of these studies are limited by their small sample size, high degree of bias, and the fact that they were conducted over a 40-year period during which the treatment for sepsis has improved and the mortality from sepsis and septic shock has decreased significantly [2,4]. The earlier studies used a short course of high-dose corticosteroid (30 mg/kg methylprednisone for up to 4 doses); this approach increased mortality and complications and was abandoned [97]. This was followed by numerous studies using a prolonged course (5–7 days) of physiologic “stress-doses” of glucocorticoids (typically 200–300 mg hydrocortisone/day). The results of these studies were mixed with some demonstrating a survival benefit, while other did not [98,99]. In 2018, two large randomized controlled trials (RCTs) were published evaluating the role of hydrocortisone in patients with septic shock [100,101]. The Activated Protein C and Corticosteroids for Human Septic Shock (APROCCHSS) study demonstrated a reduction in 90-day mortality whereas the Adjunctive Corticosteroid Treatment in Critically Ill Patients with Septic Shock (ADRENAL) study demonstrated no mortality benefit. Both studies, however, demonstrated a reduction in vasopressor dependency, duration of mechanical ventilation and ICU stay with no increased risk of complications. These studies indicate that while glucocorticoids (alone) have a biological effect in patients with septic shock, their effect on patient centered outcomes is limited. However, as indicated below, we believe that glucocorticoids act synergistically with both vitamin C and thiamine to reduce the complications and mortality associated with sepsis.

## 4. Thiamine

Thiamine is the precursor of thiamine pyrophosphate (TPP), the essential coenzyme of several decarboxylases required for glucose metabolism, the Krebs cycle, the generation of ATP, the pentose phosphate pathway, and the production of NADPH. TPP is a critical co-enzyme for the pyruvate dehydrogenase complex, the rate-limiting step in the citric acid cycle [102]. Thiamine plays an important role in many enzymatic processes involved in brain function, maintenance, and interneuronal communication [103]. Thiamine is involved in nerve tissue repair, myelin synthesis and nerve signal modulation. It plays a role in the uptake of serotonin, which in turns affects the activity of the cerebellum, the hypothalamus, and hippocampus [103]. In addition, thiamine has anti-inflammatory effects, suppressing the oxidative stress-induced activation of NF-κB [103].

Thiamine deficiency is common among septic patients, with a range in prevalence between 20% and 70%, depending on measurement techniques and inclusion criteria [104,105,106]. A deficiency in thiamine leads to decreased activity of thiamine-dependent enzymes, which triggers a sequence of metabolic events leading to energy compromise and decreased ATP production. Thiamine deficiency is associated with excitotoxic-mediated neuronal cell death [107]. Furthermore, thiamine deficiency is associated with an increased production of ROS, as well as increased expression of heme oxygenase (HO-1) and eNOS [107,108,109]. Thiamine can reverse oxidative stress that is not related to thiamine deficiency, suggesting that thiamine may act as a site-directed antioxidant [108]. It is therefore likely that thiamine deficiency compounds the oxidative mitochondrial injury and bioenergetic failure caused by vitamin C depletion.

In a pilot randomized controlled trial, Donnino et al. randomized 88 patients with septic shock to receive 200 mg thiamine twice daily for seven days [106]. In the predefined subgroup of patients with thiamine deficiency, those in the thiamine treatment group had statistically significantly lower lactate levels at 24 h and a lower mortality at 30 days. Furthermore, in a secondary analysis of this study, the need for renal replacement therapy and serum creatinine were greater in the placebo group [110]. Similarly, in a propensity matched observational study in patients with septic shock, Woolum et al. demonstrated that thiamine supplementation increased lactate clearance and decreased 28-day mortality [111].

## 5. Hydrocortisone, Ascorbic Acid, and Thiamine (HAT) in Combination

The overlapping anti-inflammatory properties of glucocorticoids and vitamin C reduce the production of pro-inflammatory mediators and ROS, which are associated with endothelial injury, mitochondrial damage, and organ failure characteristic of sepsis (see Figure 1). Furthermore, both agents have immuno-enhancing effects, which limit the immunosuppression that occurs in patients with prolonged sepsis. These agents may synergistically restore the dysregulated immune system which characterizes sepsis (see Figure 2) [112]. Thiamine may act synergistically with glucocorticoids and vitamin C to limit mitochondrial oxidative injury and restore mitochondrial function and energy production. The anti-inflammatory properties of these agents likely restore the activity of the PDC, thereby improving ATP production. However, the interaction between thiamine and ascorbic acid is complex, and likely dependent on the clinical context and ascorbic acid dosing.

The synergy between glucocorticoids and vitamin C has been established in experimental studies. Barabutis et al. have demonstrated that hydrocortisone together with vitamin C protects the vascular endothelium from damage by endotoxin, while neither agent alone had this effect [113]. Azari et al. compared the protective effects of Vitamin C (in a dose of 50 mg/kg), vitamin E and hydrocortisone alone and in combination, in a murine renal and intestinal ischemia-reperfusion model [114,115]. In these studies, both vitamin C and hydrocortisone reduced the ischemia-reperfusion injury with the combination having synergistic protective effects. In a small, retrospective, before-after-study, we demonstrated that the combination of hydrocortisone, ascorbic acid (6 g/day) and thiamine (HAT Rx) improved organ function (as reflected by the SOFA score) with a significant reduction in mortality [43]. In a similar before-after, propensity adjusted observational study, Kim et al. demonstrated a significant reduction of mortality in patients with severe pneumonia using the same treatment strategy (HAT Rx) [116]. According to the U.S. National Library of Medicine’s ClinicalTrials.gov website (https://clinicaltrials.gov/), in excess of 12 randomized controlled trials are currently underway testing vitamin C alone and in combination with hydrocortisone and thiamine in patients with severe sepsis and or septic shock. The results of these studies should provide definite information on the role of this treatment strategy in the management of patients with severe sepsis and septic shock.

## 6. Conclusions

Glucocorticoids, vitamin C and thiamine have important biological effects in patients with sepsis and septic shock. Due to the overlapping and synergistic effects of these remarkably safe and inexpensive drugs, the combination of these agents (HAT therapy) likely restores the dysregulated immune system and bioenergetic failure that characterizes sepsis. We, therefore, propose that HAT therapy will improve both the short-term (mortality) and long-term (post-sepsis syndrome) outcome of patients with sepsis and septic shock. Multiple randomized controlled trials are currently underway to test this hypothesis. 

## Figures and Tables

**Figure 1 nutrients-10-01762-f001:**
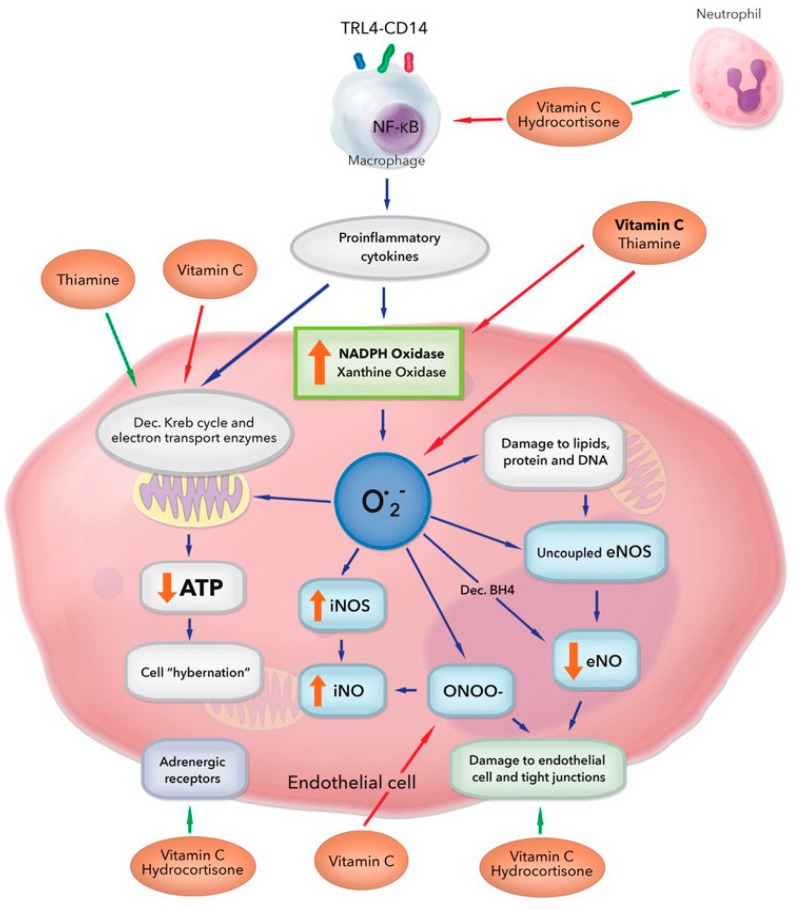
Multiple and overlapping effects of hydrocortisone, vitamin C, and thiamine in the setting of bacterial sepsis. Vitamin C and thiamine scavenge free radicals from superoxide (O_2_^−^) and inhibit activation of xanthine oxidase and NADPH oxidase. Vitamin C protects the mitochondria from oxidative stress caused by increased leakage of electrons from the dysfunctional electron transport chain and recovers tetrahydrobiopterin (BH4) from dihydrobiopterin (BH2), restoring endothelial nitric oxide synthase (eNOS) activity and increasing eNO bioavailability. Vitamin C inhibits inducible NOS (iNOS) activation, preventing profuse iNO production and peroxynitrite (ONOO^−^) generation. Vitamin C scavenges ONOO^−^, preventing loosening of the tight junctions of the endothelium. Vitamin C and hydrocortisone decrease the activation of nuclear factor κB (NF-κB), thereby decreasing the release of proinflammatory mediators. They restore endothelial tight junctions and increase adrenergic receptor function. Thiamine increases the activity of pyruvate dehydrogenase and alpha ketoglutarate dehydrogenase. These actions act in concert to restore cellular adenosine tri-phosphate (ATP) levels. ↑—increased levels/activity; ↓—decreased levels/activity. Figure adapted from Spoelstra-de Man et al. [19].

**Figure 2 nutrients-10-01762-f002:**
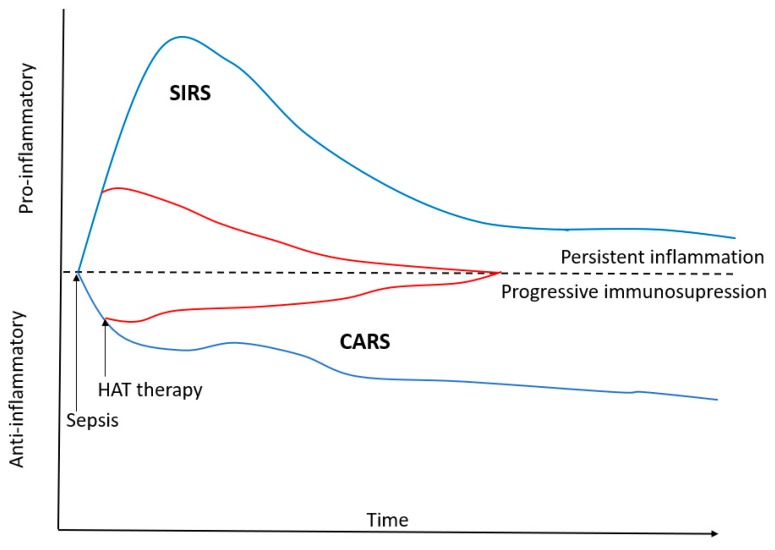
Treatment with Hydrocortisone, vitamin C and Thiamine attenuates both the pro- and anti-inflammatory response in patients with sepsis. HAT-hydrocortisone, ascorbic acid and thiamine; SIRS—Systemic Inflammatory Response Syndrome; CARS—Compensatory Anti-inflammatory Response Syndrome.

**Table 1 nutrients-10-01762-t001:** Summary of key roles of Vitamin C in sepsis.

Key Role	Mechanism
Antioxidant	Scavenges extracellular, intracellular and mitochondrial ROS; limits oxidation of mitochondrial proteins, enzymes, lipoproteins, cell membrane, etc.
Anti-inflammatory	Inhibits activation of NFκB, decreases HMGB1, inhibits histamine, prevents NETosis, inactivates HIF-1α
Microcirculation	Increases eNOS, decreases iNOS, preserves tight junctions
Immune function	Supports lymphocyte proliferation, increases neutrophil bacteriocidal action, improves chemotaxis, stimulates interferon production, decreases T regulatory cells (Tregs)
Anti-thrombotic	Decreases platelet activation and tissue factor expression, increases thrombomodulin
Synthesis of catecholamines	Acts as a cofactor in synthesis of epinephrine, dopamine and vasopressin.Increases adrenergic sensitivity
Wound Healing	Hydroxylation of procollagen, increased expression of collagen mRNA

ROS = reactive oxygen species; NFκB = nuclear factor κB; HIF-1α = hypoxia-inducible transcription factor-1α; HMGB1 = high mobility group box 1; eNOS endothelial nitric oxide synthetase; iNOS = inducible nitric oxide synthetase; HO-1 = heme oxygenase-1; HIF-1α = hypoxia-inducible transcription factor-1α2. Vitamin C: Dose response and pro-oxidant effect.

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
