# Peer review of "Hydrocortisone, Ascorbic Acid and Thiamine (HAT Therapy) for the Treatment of Sepsis. Focus on Ascorbic Acid"

_nutrients, 2018, doi:10.3390/nu10111762_

Reviewer 1 Report

Review of Nutrients-379475

This is a well-written, comprehensive and timely review of the role that vitamin C, hydrocortisone and thiamine play individually and in combination in combatting sepsis.

Major comments:

Figure 1. It is not clear from the figure legend what role vitamin C, thiamine and hydrocortisone are playing, eg what do the different coloured arrows mean – are they enhancing or inhibiting? ie this figure will need some more explanation, including definition of abbreviations.

Line 115 – vitamin C levels<11 uM are a sign of deficiency ie not necessarily ‘acute scurvy’.

Lines 110-123 – I would recommend separating out the animal (preclinical) studies/references from the human (clinical) studies/references (eg only 3 of refs 33-40 are human studies) – mainly because most of the preclinical studies have been carried out in murine models which are not necessarily a good model for the human vitamin C-requiring situation as these animals can synthesise their own vitamin C (unless a knockout) and so don’t need to uptake it through their intestines or reabsorb it via their kidneys as humans do.

Line 132 – you might want to define DHA as the two electron oxidation product of AA.

Lines 58-59 – a handful of studies have shown that vitamin C can decrease the incidence of the common cold if the person is under enhanced stress eg cold temperatures and/or physical exertion. This ties in with the association between vitamin C status and stress.

Lines 317-318 – the comment Thiamine may reduce the production of oxalate thereby reducing the risk of hyper-oxalosis associated with the administration of vitamin C. [108,109]’ is misleading because the biological pathway by which thiamine decreases oxalate formation is completely independent of ascorbate-dependent oxalate pathway. Has increased oxalate formation even been observed with IV does of 6-7 g/d vitamin C? If not, then this is a moot point.

Lines 320-321 – what is the author meaning by the statement ‘Microsomal lipid peroxidation induced by FeCl3 and ascorbate is stimulated by thiamine. [103]’? ie this may need some explanation regarding relevance.

Minor editorial comments:

Line 55 – ROS are plural

Line 57 NF-kB is singular

Line 287 – thiamine is singular

Author Response

Dear reviewer:

Thank you for your constructive comments. I have incorporated all your suggestion into  the revised manuscirpt.

Kind Regards

Paul Marik.

Reviewer 2 Report

This manuscript does a good job at providing some of the rationale behind using HAT therapy in sepsis patients. Although fairly comprehensive, it could us some corrections before publication. In addition, there are several suggested additions that could strengthen the manuscript significantly:

In general, the paragraphs are long and should be broken up. Several longer paragraphs could easily be split into two.

Figure 1: vitamin C cannot interact with peroxynitrite at any appreciable level. Also, the asorbate reaction with superoxide is slow - easy out competed by other antioxidants and native superoxide dismutase. I am unclear of the rate constant for thiamine with superoxide, but it is likely unable to compete with SOD.

Similarly, on line 131 the line "virtually all clinically important" is slightly misleading, but could just be changed to "most clinically important"

GLUT1 can only transport the oxidized form of vitamin C - dehydroascorbic acid. However a majority of the cellular ascorbate is in the reduced form. Thus, mitochondrial SVCT2 is more likely to contribute to this transport.

137-139: It is unclear what the consequences of an acute vitamin C deficiency are by way of affecting carnitine status. Several models of chronic ascorbate deficiency have showed no effects on carnitine levels. Is there any evidence this is a complication of sepsis?

Line 149: The anti-histimine effect of ascorbic acid is unclear. While some trials support this, others do not. This should be modified to add some of that uncertainty.

Line 159: Vitamin C can have effects on the common cold. The Cochrane review in question showed a statistically significant reduction in cold duration with vitamin C supplementation, it was just not of any clinically meaningful length of time. Indeed, vitamin C has much greater effects on the cold when under physical stress.

Line 178: Similar to point #5, it is unclear what an acute vitamin C deficiency will do to catecholamine levels. Several papers have shown that chronic vitamin C deficiency in some models have not altered catecholamine levels.

Line 185: Reference 68 does not provide adequate information on the transition metal reactions of ascorbate

Line 189: Reference 69 similarly does not give any information about cell culture or tissue incubation experiments.

Vitamin C plays roles in the absorption of iron and the maintenance of iron homeostasis - what roles can this play in sepsis?

Free copper can react with ascorbate much faster than iron - why was this not added to this section?

Line 283: I presume the author meant TPP vs. TTP - they are different thiamine compounds.

Line 284: thiamine is also important for the KGDC (just just the PDC); the Kreb's cycle is mentioned in the same sentence as the citric acid cycle, redundant

Thiamine plays a role in combating metabolic stress, but this is hardly touched upon here - this can easily result in the generation of ROS when thiamin is low.

The author needs to add some commentary about pre-existing thiamine and ascorbate levels vs. those as a result of septic insult. If nothing is known about this, this should be more clearly stated. 

Furthermore, does HAT therapy result in the restoration of ascorbate or thiamine levels, the maintenance of low levels of these vitamins, or the elevation of these compounds in the body past their normal physiological levels. This is important information needed to evaluate this manuscript, but could simply be stated as unanswered questions about HAT therapy.

Reference section is a little long, several could be trimmed.

Author Response

(The authors gave the same response as above.)
